# Mixed effects in machine learning – A flexible mixedML framework to add random effects to supervised machine learning regression

**Pascal Kilian**                                        *pascal.kilian@uni-tuebingen.de*
*Methods Center*
*University of Tübingen*

**Sangbeak Ye**                                         *sangbeak.ye@uni-tuebingen.de*
*Methods Center*
*University of Tübingen*

**Augustin Kelava**                                     *augustin.kelava@uni-tuebingen.de*
*Methods Center*
*University of Tübingen*

**Reviewed on OpenReview:** *https://openreview.net/forum?id=MKZyHtmfwH&layout=2&sort=date-desc*

## Abstract

Clustered data can frequently be found not only in social and behavioral sciences (e.g., multiple measurements of individuals) but also in typical machine learning problems (e.g., weather forecast in different cities, house prices in different regions). This implies dependencies for observations within one cluster, leading to violations of independent and identically distributed assumptions, biased estimates, and false inference. A typical approach to address this issue is to include random effects instead of fixed effects. We introduce the general mixed effects machine learning framework (mixedML), which includes random effects in supervised regression machine learning models, and present different estimation procedures. A segmentation of the problem allows to include random effects as an additional correction to the standard machine learning regression problem. Thus, the framework can be applied on top of the machine learning task, without the need to change the model or architecture, which distinguishes mixedML from other models in this field. With a simulation study and empirical data sets, we show that the framework produces comparable estimates to typical mixed effects frameworks in the linear case and increases the prediction quality and the gained information of the standard machine learning models in both the linear and non-linear case. Furthermore, the presented estimation procedures significantly decrease estimation time. Compared to other approaches in this area, the framework does not restrict the choice of machine learning algorithms and still includes random effects.

## 1 Introduction

Clustered data, for example, students in schools, house prices in neighborhoods or patients in hospitals are typical situations in the social and behavioral sciences, economics, and medicine (see Snijders & Bosker, 2011). It can be expected that observations within the same cluster share similarities that don't exist between observations in different clusters. If these similarities between the observations (or individuals) are not taken into account, a serious violation of the independent and identically distributed (i.i.d.) assumption has to be expected, leading to biased estimates, false inferences, etc. Throughout this paper, we use the example of predicting a student's test score. After including information on the student level (level 1), like age or cognitive ability, the prediction residuals can be seen as a result of unobserved information

about the students. These residuals are assumed to be i.i.d., but within-school (cluster) similarities violate the independence assumption. The within-school dependency can be described by school-level (level 2) information like average cognitive abilities. On this level, unobserved information (like class sizes) might also influence prediction results.

In order to account for the dependencies of clustered data, one way is to introduce fixed effects (e.g., dummy variables/one-hot encoding, which describe the membership to a data cluster beyond the available cluster information). However, this leads to a rapid increase in the number of parameters and to a loss of degrees of freedom. In the past, another way to account for the dependencies has been to include random effects. In this scenario, for each cluster, a random draw from a parametric distribution (i.e., normal distribution) is conducted, which represents the cluster effect in the measurement. In other words, each observed value is decomposed into a part which is shared by all observational units for the same cluster and another part that represents the deviation of an observational unit from the cluster (mean). The random effects approach implies that only those parameters are estimated which belong to the parametric distribution (e.g., one mean and one distributional parameter for the normal distribution). As a result, the loss of degrees of freedom is limited. In the school example, this means on level 2, we use the observed school information (e.g. average cognitive abilities) as fixed effects that influence the students' test score prediction. The effect of this school information is the same for all students (dependent on the average cognitive abilities of their respective schools). But for each school, a random draw from a parametric distribution (as a realization of a random variable) describes the deviation of the expected school influence on the prediction (accounting, for example, for different class sizes / student-teacher ratios that are not observed but are expected to influence test scores). In this sense, it is also helpful to think about the specific set of schools, included in the data, as a random subset of all possible schools.

In this paper, we will describe how random effects can be included in the more general setting of machine learning algorithms. In contrast to self-contained models, we present a framework that can take existing models (without random effects) and extend them (by fine-tuning) by random effects. We will also discuss and introduce different estimation procedures for the variance components of this framework. In the next sections, we introduce the significant related work in this area and describe the contribution of our framework to the existing literature. In section 2 we introduce mixed effects models and the notation for the paper. In section 3 we describe the proposed framework followed by the estimation in section 4. Following the properties of a real data set, we use simulated data for evaluation in section 5 and apply the framework on real data in section 6.

**Related work**  The idea of adding random effects to prediction algorithms is, in principle, known. However, previous research mostly focused on tree-based methods for the (non-linear) prediction of the fixed effects part of the problem. Fokkema et al. (2018) presented the (generalized) linear mixed effects model trees (GLMM tree). Similar results can be found as (generalized) mixed effects regression trees or random forests in Hajjem et al. (2011; 2014; 2017) or Sela & Simonoff (2012). Also, Nestler & Humberg (2021) proposed a cross-sectional regression tree and added regularization (LASSO) and more random components. Most of these approaches are based on an Expectation-Maximization (EM) procedure, where standard mixed model frameworks are incorporated (from here on referred to as out-of-the-box - OOB implementation). This approach is widely known as the Random Effect - Expectation-Maximization (RE-EM) tree from (Sela & Simonoff, 2012). Some frameworks (e.g. Hajjem et al., 2017; Xiong et al., 2019) did not use OOB functions, but rely on Maximum Likelihood Estimation (MLE)-based RE-EM. In the framework *Mixed effect machine learning* (MeML, Ngufor et al., 2019) a more general approach is suggested, where (in theory) the choice of the machine learning model for the fixed effect prediction is free, and, for the random effects, a GLMM model is used. While they pointed out that, in theory, all kinds of machine learning models can be used, the implementation and application still is restricted to tree-based models (with an indicator variable) and OOB frameworks for the random part. The same ideas can be found in Xiong et al. (2019). Here, a deep learning architecture (from the field of computer vision) was applied to receive representations of images. For those representations, an EM algorithm is used to estimate the mixed effects model. Mandel et al. (2022) developed a model which substitutes the linear predictor in GLMMs with a feed-forward neural network and therefore bears the highest resemblance to the mixed effects machine learning (mixedML) framework described here. Using simulated data and a data set to predict depression and anxiety levels of schizophrenia

they showed the advantages of their model. We refer to this paper for a very thorough derivation of a model that combines machine learning and mixed-effect modeling. While we absolutely recommend the recent research results for application and as a presentation of different models in this area, we show with this paper that random effects can be considered on the basis of a broad range of existing models, even after they have been trained. In this sense, this framework is a generalization of the above results that includes most of the introduced methods. Therefore, in this paper, we do not aim to introduce a new model that competes with the mentioned research. Instead, we unify the methods in one framework and add new estimation methods within the framework as well as provide more flexibility for the choice of the ml approach.

**Contributions** Our framework differs from the mentioned literature in the flexibility of the machine learning regression procedure. This is because we are not presenting a model that combines a specific machine learning algorithm with random effects, but rather a framework that uses a placeholder for the model that is used for the fixed effects part. Further, our framework merges and complements various estimation methods presented previously.

Therefore, our contribution to the literature is three-fold. First, we combine different approaches into one framework. Here, we refer to the estimation procedures. In general, and also in our framework, these are based on the introduced RE-EM idea (which was applied as RE-EM trees to tree-based models). As presented, both MLE (similar to Hajjem et al., 2014; Xiong et al., 2019) and OOB methods were introduced within this approach. Besides these methods, our framework also provides a Restricted Maximum Likelihood estimation (REML) which is a common approach to estimating variance and covariance components in mixed effects models. In a simulation study, those estimations procedures are compared, especially with respect to computing time which decreases significantly for the MLE/REML procedures compared to OOB.

Second, our framework provides great flexibility in the estimation of the fixed effects. The framework can directly be applied to a wide range of supervised regression machine or deep learning models and thus increasing their predictive power (Sela & Simonoff, 2012). A very important aspect here is how the models are trained within each EM iteration. The models only need to be adjusted with respect to a slightly changed target variable in each iteration (comparable to transfer learning). This means that a trained model only needs to be fine-tuned in each iteration, which is a considerable time-saving when using model-dependent convergence criteria. From this, another enormous advantage of the framework can be derived. While the framework allows to train a model from scratch, an already specified and trained model can be used and adjusted under consideration of random effects for the reason described above.

Third, we use the simulation and real data to access the benefits of the framework in the sense of model specification. Here we show that for prediction quality, it is not necessary to specify non-linearities like interactions in the model (the traditional way in psychometrics) as long as the machine learning procedure is capable of modeling the non-linear relationship.

Through the above points, we complement the existing methods. Specified as special cases, the framework can also serve as a summary of the mentioned research (e.g. tree-based methods can still be used as specification). Taken together, these enhancements will enable researchers to investigate a possible prediction improvement in the mixed model domain (under consideration of different implemented estimation procedures) after having completed the analysis in the machine learning domain, using all the time and work previously invested (hyperparameter tuning, training of the model, ...). In that sense, the framework can be applied on top of existing models and analyses. With this, the advantages of the usage of mixed effects become available to a broad range of machine learning research.

## 2 Mixed effects regression models

### 2.1 Linear Mixed Effects (LME) Model

Using the notation of Laird & Ware (1982) or Demidenko (1987), the traditional mixed effects regression model can be described as follows:

$$\boldsymbol{y}_j = \boldsymbol{X}_j\boldsymbol{\beta} + \boldsymbol{Z}_j\boldsymbol{\nu_j} + \boldsymbol{\epsilon_j}, \ j \in [1, ..., J], \tag{1}$$

where $\boldsymbol{y}_j$ is the $(n_j \times 1)$ vector of observed dependent variables (vector of test scores of the $n_j$ students in school $j$, from the introductory example) for each cluster $j$ ($n_j$ is the sample size within a cluster $j$), $\boldsymbol{X}_j$ is the $(n_j \times p)$ matrix of $p$ observed variables of $n_j$ observational units, $\boldsymbol{\beta}$ is the $(p \times 1)$ dimensional vector of unknown regression coefficients (parameters), $\boldsymbol{Z}_j$ is a $(n_j \times p_r)$ design matrix, $\boldsymbol{\nu_j}$ is a $(p_r \times 1)$ vector of unobserved random variables (random effects) with $\boldsymbol{\nu}_j \sim \mathcal{N}(\boldsymbol{0}_{p_r}, \boldsymbol{\Sigma})$, and $\boldsymbol{\epsilon}_j$ is the $(n_j \times 1)$ vector of normally distributed residual variables with $\boldsymbol{\epsilon}_j \sim \mathcal{N}(\boldsymbol{0}_{n_j}, \sigma_\epsilon^2 \boldsymbol{I}_{n_j})$.

Note that in $\boldsymbol{X}_j$, we collect the level 1 and level 2 covariates as columns (observed student (level 1) and school (level 2) information). In other words, the first $p_1$ columns describe the observed covariates of the $n_j$ observational units from cluster $j$, and the remaining $p_2$ columns describe the observed covariates which give information on the cluster $j$ (being constant for the $n_j$ observational units form cluster $j$). $p_r$ is the number of observed covariates for which random effects are included. Without loss of generality and for the ease of notion, we will assume that the covariates are ordered, so that the first $p_r$ covariates are assumed to have cluster-dependent (random) effects.

Furthermore, we use the notation $\hat{\boldsymbol{y}}_j = \boldsymbol{X}_j \hat{\boldsymbol{\beta}} + \boldsymbol{Z}_j \hat{\boldsymbol{u}}_j$ for expected values and define for an individual observation $i$ of the cluster $j$:

$$\hat{y}_{ij} = \hat{y}_{ij}^{\text{fixed}} + \boldsymbol{z}_{ij} \hat{\boldsymbol{u}}_j, \tag{2}$$

with the respective row $\boldsymbol{z}_{ij} = (1, \tilde{\boldsymbol{x}}_{ij})$ in the design matrix $\boldsymbol{Z}_j$, consisting of a bias/intercept term ($= 1$) for the random intercept and the feature vector $\tilde{\boldsymbol{x}}_{ij} = \boldsymbol{x}_{ij}[1 : p_r]$. The estimated random effects (i.e., realizations of the latent/random variables) are $\hat{\boldsymbol{u}} = (\hat{\boldsymbol{u}}_1', ..., \hat{\boldsymbol{u}}_J')'$ with $\hat{\boldsymbol{u}}_j = (\hat{u}_0, ..., \hat{u}_{p_{rand}})'$ for cluster $j$. Then $\boldsymbol{y}_j$ is conditionally normal distributed

$$\boldsymbol{y}_j \sim \mathcal{N}(\boldsymbol{y}_j^{\text{fixed}}, \boldsymbol{V}_j), \tag{3}$$

$$\boldsymbol{V}_j = \boldsymbol{Z}_j \boldsymbol{\Sigma} \boldsymbol{Z}_j' + \sigma_\epsilon^2 \boldsymbol{I}_{n_j}. \tag{4}$$

In matrix notation we can write (over all $J$ clusters): $\boldsymbol{y} = \boldsymbol{X}\boldsymbol{\beta} + \boldsymbol{Z}\boldsymbol{\nu} + \boldsymbol{\epsilon}$ with $\boldsymbol{\nu} = (\boldsymbol{\nu}_1', ..., \boldsymbol{\nu}_J')'$ and the design block matrix

$$Z = \begin{pmatrix} Z_1 & ... & 0 \\ ... & ... & ... \\ 0 & ... & Z_J \end{pmatrix}. \tag{5}$$

Variance and covariance components in LMEs are typically estimated by the EM algorithm, based on either MLE or REML. As elaborated in Raudenbush & Bryk (2002), MLE estimates are asymptotically consistent and efficient. However, for small sample sizes on level 2 (i.e., number of clusters; which is not the case in our paper), MLE estimates of the variances and covariances of the random effects will be underestimated. In order to avoid this issue, REML estimates can be obtained which give unbiased estimates and take into account the unreliability of the fixed effects. Again, we would like to emphasize that this is an issue for small sample sizes given in social and behavioral sciences (e.g., with a number of clusters $< 30$). As a result of this, in practical situations estimation of the random effects variances and covariances relies on REML, and estimation of fixed effects relies on MLE in so-called multilevel modeling.

## 2.2 Non-linear mixed effects models

As a generalization of the linear mixed effects model, the non-linear mixed effects model specifies the fixed effects term as a non-linear function of the population parameter $\boldsymbol{\beta}$ as follows:

$$\boldsymbol{y}_j = \boldsymbol{f}_\beta(\boldsymbol{X}_j) + \boldsymbol{Z}_j \boldsymbol{\nu}_j + \boldsymbol{\epsilon}_j, j \in [1, ..., J]. \tag{6}$$

As a consequence, we have a closed-form analytic expression for the marginal mean and covariance terms of the model, $E[\boldsymbol{y}_j] = \hat{\boldsymbol{f}}_\beta(\boldsymbol{X}_j)$ (following the conventional notation of non-linear regression model).

Due to these properties, this model is referred to as non-linear marginal models Demidenko (1987) or as population-averaged models (Liang & Zeger (1986); Zeger & Liang (1986)), because the marginal function of the response variable is expressed as a function of population-level covariates. This specification differs from the fully non-linear mixed effects model in that the random effects are not included in the non-linear

kernel, wherein this model is also called the partially non-linear mixed effects model (Davidian & Giltinan (1995)). In non-linear mixed effects models, estimation of the models relies on the marginal distribution of the response variable. However, as we assume an explicit expression of the distribution of the mean and covariance structure of the response variable, there is no need for integration.

The model for the dependent variable then takes a conditional form, $\boldsymbol{y}_j|(\boldsymbol{X}_j, \boldsymbol{Z}_j, \boldsymbol{\nu}_j) = \boldsymbol{f}_{\boldsymbol{\beta}}(\boldsymbol{X}_j) + \boldsymbol{Z}_j\boldsymbol{\nu}_j + \boldsymbol{\epsilon}_j$. Without loss of generality, we assume that $\boldsymbol{\epsilon}_j$ and $\boldsymbol{\nu}_j$ are normally distributed. The non-linear mixed effects model can be written as the following:

$$\boldsymbol{y}_j \sim \mathcal{N}(\boldsymbol{f}_{\boldsymbol{\beta}}(\boldsymbol{X}_j), \boldsymbol{Z}_j\boldsymbol{\Sigma}\boldsymbol{Z}_j' + \sigma_\epsilon^2\boldsymbol{I}_{n_j}), \tag{7}$$

and $\boldsymbol{f}_{\boldsymbol{\beta}}(\boldsymbol{X}_j) = \boldsymbol{y}_j^{\text{fixed}}$ as before.

## 3   Mixed effects models for machine learning

While the fixed effects approach can be implemented straightforwardly using one-hot encoded variables for the cluster membership, it is not obvious how to implement the random effects. In the next section, we introduce the framework mixedML to implement the random effects.

### 3.1   Mixed effects approach - mixedML

If the data contains features on both the observation level (level 1) $\boldsymbol{X}_1 \in \mathbb{R}^{N \times p_1}$ and the cluster level (level 2) $\boldsymbol{X}_2 \in \mathbb{R}^{J \times p_2}$ we include the level 2 features with fixed effects in the level 1 prediction. For that we combine the matrices to obtain $\boldsymbol{X} = [\boldsymbol{X}_1, \tilde{\boldsymbol{X}}_2]$ with $\tilde{\boldsymbol{X}}_2 \in \mathbb{R}^{N \times p_2}$, where row $i$ of $\tilde{\boldsymbol{X}}_2$ is row $j$ of $\boldsymbol{X}_2$ for all $i$ in cluster $j$. Note that this does not correspond directly to a mixed model with level two covariates since the level 2 covariates only determine the cluster deviations from the overall intercept, instead of influencing the cluster-specific slopes. In our model, the cluster-specific slopes are only determined by the centered random effect. From here on $X$ is seen as a feature matrix that might contain level 2 features, which are constant across observations within the same cluster.

Instead of estimating a linear relationship between $\boldsymbol{X}$ and $\boldsymbol{y}$ via $\boldsymbol{X}\boldsymbol{\beta}$ for the fixed effects, we use a potentially non-linear function $f(\boldsymbol{X})$ to model this relationship via $\boldsymbol{y} = f(\boldsymbol{X}) + \boldsymbol{Z}\boldsymbol{\nu} + \boldsymbol{\epsilon}$ and then add the linear random effects with the same assumptions as in section 2.

To parameterize the function $f$, we use a machine learning model, referred to as $\text{ml}_{fixed(\theta)}$ and add the random effects (cluster differences), which can be seen as a linear *correction*. This results in

$$\boldsymbol{y} = \text{ml}_{\text{fixed}(\boldsymbol{\theta})}(\boldsymbol{X}) + \boldsymbol{Z}\boldsymbol{\nu} + \boldsymbol{\epsilon}, \tag{8}$$

or in the cluster specific notation (compare to (1)) $\hat{\boldsymbol{y}}_j = \text{ml}_{\text{fixed}(\boldsymbol{\theta})}(\boldsymbol{X}_j) + \boldsymbol{Z}_j\hat{\boldsymbol{u}}_j$. For the estimation we split $\boldsymbol{y}$ into a fixed and random part $\boldsymbol{y} = \boldsymbol{y}^{\text{fixed}} + \boldsymbol{y}^{\text{rand}} + \boldsymbol{\epsilon}$ where $\boldsymbol{y}^{\text{fixed}} = \boldsymbol{y} - \boldsymbol{Z}\boldsymbol{u}$ and $\boldsymbol{y}^{\text{rand}} = \boldsymbol{e}^{\text{fixed}} = \boldsymbol{y} - \boldsymbol{y}^{\text{fixed}}$, with the cluster specific realizations $\boldsymbol{u} = (\boldsymbol{u}_1', ..., \boldsymbol{u}_J')'$ of the random effects/latent variables $\boldsymbol{\nu}$.

Before we describe further details in the next section, we give a general idea, following related ideas for tree-based models (e.g. Hajjem et al., 2011; 2014; Sela & Simonoff, 2012) and Hajjem et al. (2011); Xiong et al. (2019). We train $\text{ml}_{\text{fixed}(\theta)}$ by ignoring cluster effects (with the target $\boldsymbol{y}$) to get a first estimate for $\boldsymbol{y}^{\text{fixed}}$ (the prediction $\hat{\boldsymbol{y}}^{\text{fixed}}$ of the model). With this we estimate $\hat{\boldsymbol{u}}$ based on $\boldsymbol{e}^{\text{fixed}}$. Then we update $\boldsymbol{y}^{\text{fixed}} = \boldsymbol{y} - \boldsymbol{Z}\hat{\boldsymbol{u}}$ using the current estimate $\hat{\boldsymbol{u}}$ and re-train $\text{ml}_{\text{fixed}(\theta)}$ with the updated target variable $\boldsymbol{y}^{\text{fixed}}$. This process is iterated until some convergence criteria are met or the maximal number of iterations is reached. Since, for every iteration, a (potentially complex) machine learning algorithm needs to be trained, the question of training time arises. To provide some clarity here, in each iteration the model is re-used and fine-tuned to an adjusted prediction task. In the first iteration, we train the model for a task $(\boldsymbol{X}, \boldsymbol{y})$. In the subsequent iterations, we update the trained network of the last iteration and tune the parameters for the updated task $(\boldsymbol{X}, \boldsymbol{y}^{\text{fixed}})$ (comparable to transfer learning).

## 4 Estimation

To get an estimate for $\boldsymbol{y}^{\text{fixed}}$ we learn the parameters $\hat{\boldsymbol{\theta}}$ of $\texttt{ml}_{\text{fixed}(\theta)}$ (e.g. by SGD methods) and predict $\hat{\boldsymbol{y}}^{\text{fixed}} = \texttt{ml}_{\text{fixed}(\hat{\theta})}(\boldsymbol{X})$. If $\boldsymbol{u}$ is known, common supervised machine learning regression methods can be trained and applied to predict $\boldsymbol{y}^{\text{fixed}}$, including the special case of $\boldsymbol{u} = 0$ (and thus $\boldsymbol{y}^{\text{fixed}} = \boldsymbol{y}$) where random effects are ignored (typical case). Since the random part of the model (and therefore $\boldsymbol{u}$) is unknown, we need to find estimates $\hat{\boldsymbol{u}}$ to use in (3.1) to define the machine learning task. But to get an estimate for the random part of the model, we need the estimate $\hat{\boldsymbol{y}}^{\text{fixed}}$ for (3.1). This motivates an alternating, iterative process like the EM-algorithm where estimates of the current fixed or random parameters are used for the respective updates (see e.g. Sela & Simonoff, 2012; Hajjem et al., 2014; Xiong et al., 2019).

### 4.1 EM

Following common techniques for the EM-algorithm (Lindstrom & Bates, 1988; Wu & Zhang, 2006; Laird & Ware, 1982) (similar as in Hajjem et al. (2011)), we can use the EM algorithm as an estimator for the variance components $\sigma$ and $\boldsymbol{\Sigma}$ (for both, MLE and REML). MLE and REML estimates can be obtained by using the respective projections $\boldsymbol{P}_{j\text{MLE}} = \boldsymbol{V}_j^{-1}$ and $\boldsymbol{P}_{j\text{REML}} = \boldsymbol{V}_j^{-1} - \boldsymbol{V}_j^{-1}\boldsymbol{X}_j(\boldsymbol{X}^T\boldsymbol{V}^{-1}\boldsymbol{X})^{-1}\boldsymbol{X}_j^T\boldsymbol{V}_j^{-1}$ (see e.g. Lindstrom & Bates, 1988), where $\boldsymbol{V}_j$ is defined as in (4). Given the estimates $\sigma_\epsilon^{(t-1)}$, $\boldsymbol{\Sigma}^{(t-1)}$ and $\hat{\boldsymbol{u}}_j^{(t-1)}$ from the last iteration $(t-1)$, we can (re)train $\texttt{ml}_{\text{fixed}}^{(t)}(\boldsymbol{X}_j)$ using the target $\boldsymbol{y}_j^{\text{fixed}} = \boldsymbol{y}_j - \boldsymbol{Z}_j\hat{\boldsymbol{u}}_j^{(t-1)}$ and calculate $\hat{\boldsymbol{b}}_j^{(t)}, \boldsymbol{r}_j^{(t)}$ and $\boldsymbol{V}_j^{(t-1)}$ (evaluated at $\boldsymbol{\Sigma}^{(t-1)}$) as

$$\boldsymbol{V}_j^{(t-1)} = \boldsymbol{Z}_j\boldsymbol{\Sigma}^{(t-1)}\boldsymbol{Z}_j^T + \sigma^2\boldsymbol{I},$$
$$\hat{\boldsymbol{u}}_j^{(t)} = \boldsymbol{\Sigma}^{(t-1)}\boldsymbol{Z}_j^T\boldsymbol{V}_j^{(t-1)-1}\left(\boldsymbol{y}_j - \texttt{ml}_{\text{fixed}}^{(t)}(\boldsymbol{X}_j)\right),$$
$$\boldsymbol{r}_j^{(t)} = \boldsymbol{y}_j - \texttt{ml}_{\text{fixed}}^{(t)}(\boldsymbol{X}_j) - \boldsymbol{Z}_j\hat{\boldsymbol{u}}_j^{(t)}.$$

The new estimates $\sigma_{\epsilon\text{MLE/REML}}^{(t+1)}$ and $\boldsymbol{\Sigma}_{\text{MLE/REML}}^{(t+1)}$ are calculated as

$$\boldsymbol{\Sigma}_{\text{MLE/REML}}^{(t)} = \frac{1}{J}\sum_{j=1}^J\left[\hat{\boldsymbol{u}}_j^{(t)}\hat{\boldsymbol{u}}_j^{(t)T} + \left[\boldsymbol{\Sigma}^{(t-1)} - \boldsymbol{\Sigma}^{(t-1)}\boldsymbol{Z}_j^T\boldsymbol{P}_{j\text{MLE/REML}}^{(t-1)}\boldsymbol{Z}_j\boldsymbol{\Sigma}^{(t-1)}\right]\right],$$

$$\left(\sigma_{\epsilon\text{MLE/REML}}^{(t)}\right)^2 = \frac{1}{N}\sum_{j=1}^J\left[\boldsymbol{r}_j^{(t)T}\boldsymbol{r}_j^{(t)} + \sigma_\epsilon^{2(t-1)}\text{trace}\left(\boldsymbol{I} - \sigma_\epsilon^{2(t-1)}\boldsymbol{P}_{j\text{MLE/REML}}^{(t-1)}\right)\right].$$

Note that instead of inverting $\boldsymbol{V}_j, \forall j = 1, \ldots, J$ we can invert $\boldsymbol{\Sigma}$ and use $\boldsymbol{V}_j^{-1} = \boldsymbol{I} - \boldsymbol{Z}_j\left(\boldsymbol{Z}_j^T\boldsymbol{Z}_j + \boldsymbol{\Sigma}^{-1}\right)^{-1}\boldsymbol{Z}_j^T$ where $\left(\boldsymbol{Z}_j^T\boldsymbol{Z}_j + \boldsymbol{\Sigma}^{-1}\right)$ is usually smaller (number of random effects squared) than $\boldsymbol{V}_j$ $(n_j \times n_j)$.

Convergence is monitored by the generalized log-likelihood (GLL)

$$GLL(\boldsymbol{\theta}, \boldsymbol{b}_j|\boldsymbol{y}_j) = \sum_{j=1}^J\left[\boldsymbol{r}_j^T(\sigma_\epsilon^2\boldsymbol{I})\boldsymbol{r}_j + \boldsymbol{u}_j^T\boldsymbol{\Sigma}^{-1}\boldsymbol{u}_j + \log|\boldsymbol{\Sigma}| + \log|\sigma_\epsilon^2\boldsymbol{I}|\right]. \tag{9}$$

We implemented this approach and presented the algorithm in Table 1. Note that the separated estimation of random effects and fixed effects in each iteration $t$ gives great flexibility for both parts. For the fixed effects, we can implement any supervised regression machine learning algorithm that learns to predict $\boldsymbol{y}^{\text{fixed}}$ and produces the estimate/prediction $\hat{\boldsymbol{y}}^{\text{fixed}}$. For the random effects, we can also use standard LME methods (e.g. MixedLM() from the statsmodels module (Seabold & Perktold, 2010) in python, or lmer() from lme4 (Bates et al., 2015) package in R) where the dependent/target variable is defined as $\boldsymbol{y}^{\text{rand}(t)} = \boldsymbol{y} - \hat{\boldsymbol{y}}^{\text{fixed}(t)}$ in iteration $t$. As mentioned before, we refer to this implementation as OOB (see Appendix A for the algorithm).

Table 1: Implementation of the MLE/REML based EM estimation.

| Algorithm |
|---|

| $t = 0$ | Initialize $\hat{\boldsymbol{u}}_{j(0)} = 0, \forall j = 1, ..., J$, $\boldsymbol{y}_{(0)}^{\text{fixed}} = \boldsymbol{y}$, $\hat{\boldsymbol{\Sigma}}_{\nu(0)} = \boldsymbol{I}$, $\hat{\sigma}_{\epsilon(0)}^2 = 1$ and train $\texttt{ml}_{\text{fixed}(\theta)}(\boldsymbol{X}) = \boldsymbol{y}_{(0)}^{\text{fixed}} \rightarrow \hat{\boldsymbol{y}}_{(0)}^{\text{fixed}}$ |
|---|---|
| $t - 1 \rightarrow t$ | for all $j$ |

$$V_j^{t-1} = \boldsymbol{Z}_j \boldsymbol{\Sigma}^{(t-1)} \boldsymbol{Z}_j^T + \sigma^2 \boldsymbol{I}$$

$$\hat{\boldsymbol{u}}_j^{(t)} = \boldsymbol{\Sigma}^{(t-1)} \boldsymbol{Z}_j^T V_j^{(t-1)-1} \left( \boldsymbol{y}_j - \texttt{ml}_{\text{fixed}}^{(t)}(\boldsymbol{X}_j) \right)$$

$$\boldsymbol{r}_j^{(t)} = \boldsymbol{y}_j - \texttt{ml}_{\text{fixed}}^{(t)}(\boldsymbol{X}_j) - \boldsymbol{Z}_j \hat{\boldsymbol{u}}_j^{(t)}$$

**MLE:** $\boldsymbol{P}_{j\text{MLE}} = \boldsymbol{V}_j^{-1}$

**REML:** $\boldsymbol{P}_{j\text{REML}} = \boldsymbol{V}_j^{-1} - \boldsymbol{V}_j^{-1} \boldsymbol{X}_j (\boldsymbol{X}^T \boldsymbol{V}^{-1} \boldsymbol{X})^{-1} \boldsymbol{X}_j^T \boldsymbol{V}_j^{-1}$

$$\boldsymbol{\Sigma}_{\text{MLE/REML}}^{(t)} = \frac{1}{J} \sum_{j=1}^J \left[ \hat{\boldsymbol{u}}_j^{(t)} \hat{\boldsymbol{u}}_j^{(t)T} + \left[ \boldsymbol{\Sigma}^{(t-1)} - \boldsymbol{\Sigma}^{(t-1)} \boldsymbol{Z}_j^T \boldsymbol{P}_{j\text{MLE/REML}}^{(t-1)} \boldsymbol{Z}_j \boldsymbol{\Sigma}^{(t-1)} \right] \right]$$

$$\left( \sigma_{\epsilon\text{MLE/REML}}^{(t)} \right)^2 = \frac{1}{N} \sum_{j=1}^J \left[ \boldsymbol{r}_j^{(t)T} \boldsymbol{r}_j^{(t)} + \sigma_\epsilon^{2(t-1)} \text{trace} \left( \boldsymbol{I} - \sigma_\epsilon^{2(t-1)} \boldsymbol{P}_{j\text{MLE/REML}}^{(t-1)} \right) \right]$$

$$\boldsymbol{y}_{(t)}^{\text{fixed}} = \boldsymbol{y} - \boldsymbol{Z} \hat{\boldsymbol{u}}_{(t)}$$

train $\texttt{ml}_{\text{fixed}(\theta)}(\boldsymbol{X}) = \boldsymbol{y}_{(t)}^{\text{fixed}} \rightarrow \hat{\boldsymbol{y}}_{(t)}^{\text{fixed}}$

Note that both algorithms include an isolated and complete training (using convergence criteria) of the machine learning procedure (ignoring random effects) in the first part of the first iteration. This provides great flexibility in the choice of the machine learning procedure and the respective training process. Within the optimization, the machine learning model can be learned from scratch, but the process can also be started with pre-trained models, which leads to early stops according to convergence criteria. In this case, the existing pre-trained model will be fine-tuned to consider random effects. In every subsequent iteration, parameters will be updated according to updated expectations of the random effect. We will refer to those three estimation methods as MLE, REML, and OOB.

### 4.2 Prediction

The problem of predicting an unseen data point can be divided into two tasks. In the first task, we want to predict the outcome $y_{i'j}$ of a new observation $i'$ in the cluster $j$, and the cluster is known from the training data ($j \in [1, ..., J]$). We call this task *within sample prediction*. In the second task, we predict a new outcome where we have information about the cluster (the level 2 features), but the cluster itself might not be one of the clusters considered during training. We refer to this task as *out of sample prediction*. The predictions are set to

$$y_{i'j}^{\text{out}} = \hat{y}_{i'j}^{\text{fixed}} = \texttt{ml}_{\text{fixed}(\theta)}(X_{i'j}), \tag{10}$$

$$y_{i'j}^{\text{within}} = \hat{y}_{i'j} = \texttt{ml}_{\text{fixed}(\theta)}(X_{i'j}) + \boldsymbol{Z}\hat{\boldsymbol{u}}_j, \tag{11}$$

where $\boldsymbol{Z}$ is specified as above and $\hat{\boldsymbol{u}}_j$ is the realisation of $\boldsymbol{\nu}_j$ for the known cluster $j$. Note that we use the very simple approach of using the information of random effects if it is present. Random effects could also be predicted for unknown clusters as in Xiong et al. (2019). For simplicity of the framework introduction, we have refrained from including this in the paper to focus on model comparisons with and without the consideration of random effects. In the test score example, the *within sample prediction* is the test score prediction of a student in a school that was included in the training data. Here we know the schools' realisation of the random effects following the population (of possible schools) distribution and we can use this information. In the *out of sample prediction* we only know the observed information of the student's school and thus can only use the common fixed effects.

## 5 Simulation studies

To evaluate and compare the models, we conducted a simulation study. The simulated data sets follow the properties of a real data set from social sciences. We give the relevant settings in this section. Further details

can be found in Appendix B. For sample sizes we used $J = 200$ with $n_j$ sampled between 22 and 25 (giving $N = 5707$). The dimension of the feature space is $p_1 = 10$, $p_2 = 3$, and includes random effects for all level 1 predictors ($p_r = p_1 + \text{intercept/bias}$).

For the covariance matrix $\Sigma_\nu$ of the $p_r + 1$ random effects (level 2 latent variables) we assume $\sigma_{\nu_m, \nu_n} = 0$ for $n \neq m$ and set $\sigma^2_{\nu_m} = 1, \forall m \in [0, 1, ..., p_r])$. By setting the variance of the random effects to one, we identify the latent variable vector.

Cluster specific realizations $\boldsymbol{u}_j$ of the latent variables are sampled as $\boldsymbol{u}_j \sim \mathcal{N}(\boldsymbol{0}, \Sigma_\nu)$ for $j = 1, ..., J$.

For the residuals we assume $\Sigma_\epsilon = \sigma_\epsilon \boldsymbol{I}$ with $\sigma_\epsilon = 3.5$, and sample $\boldsymbol{\epsilon} \sim \mathcal{N}(\boldsymbol{0}, \Sigma_\epsilon)$ ($\boldsymbol{\epsilon} \in \mathbb{R}^N$)

We set the intercept/bias $\beta_0 = 5$ and all other parameters to 1 and obtain $\boldsymbol{\beta}_1 \in \mathbb{R}^{p_1}$ (level 1 parameters/weights), $\boldsymbol{\beta}_2 \in \mathbb{R}^{p_2}$ (level 2 parameters/weights). With the chosen values, together with $\sigma_\epsilon$, we achieve an overall explained variances of $y$ of around 0.77 (linear) and 0.82 (non-linear), which would already be considered as high but plausible (at least for repeated measures) in the social sciences.

To show the downward compatibility of the framework, we simulate the linear case by $\boldsymbol{y}_{\text{lin}} = \boldsymbol{\beta}_0 + \boldsymbol{X}\boldsymbol{\beta} + \boldsymbol{Z}\boldsymbol{u} + \boldsymbol{\epsilon}$. To show the benefit of more complex relationships, we simulate the non-linear data set by including quadratic terms of the first five level 1 features and define $\boldsymbol{X}_q = (x_{11}^2, ..., x_{15}^2)$ and $\boldsymbol{\beta}_q \in \mathbb{R}^5$ sampled as the other coefficients. This gives

$$\boldsymbol{y}_{\text{nlin}} = \boldsymbol{\beta}_0 + \boldsymbol{X}\boldsymbol{\beta} + \boldsymbol{X}_q\boldsymbol{\beta}_q + \boldsymbol{Z}\boldsymbol{u} + \boldsymbol{\epsilon}. \tag{12}$$

## 5.1 Evaluation and comparison metrics

In the analysis, we always specify and train the machine learning algorithm first and report the results (`ml`). As a fixed effects comparison, we specify the same model but include the one-hot encoded cluster-membership as predictor (`hot`). We then use the respective `ml` model within the mixedML framework and train it with all introduced estimation methods (MLE, REML and OOB). Following this procedure from here on, we only introduce the `ml` method on which the other models are based on.

In the evaluation and comparison of the framework, we focus on two aspects. First, we compare models with regard to parameter estimation and estimation time. Second, we compare the prediction qualities for new in-sample observations (*within* - the cluster of the observations is known from the training data) and out-of-sample observations (*out* - all information is given, but it is unknown if the cluster is one of the training clusters). For the prediction quality, we use the root-mean-squared-error (RMSE). Again, in the test score example, an in-sample observation (*within*) refers to a student who goes to a school that is known from training (and thus the respective estimated mean deviation can be used). In an out-of-sample observation, the student's school is one of all possible schools, and apart from the common fixed effects we have no further information. The differences in the according RMSE values are according to whether or not this information can be used. Note that the `ml` and `hot` models don't include the random effects. This means the *within* and *out* scores are the same for the `ml` model (a within-sample observation does not provide further information). For the `hot` there is no real *out* score because only known clusters are represented in the encoding.

## 5.2 Models and procedure

In this paper, the focus is to compare approaches regarding the consideration of random effects. In order to perform the comparisons on reasonable machine learning models, hyperparameters for those models are tuned separately (for details, see Appendix C). The same models (with the found specifications) then are used as pure machine learning models (`ml`) and within the mixedML framework.

In the linear case, we specify the `ml` model to be a linear regressor and use a linear feed-forward neural network ($\text{NN}_{\text{lin}}$ - no hidden layer) and compare it to a traditional LME model (statsmodels), specified the same way. Note that $\text{NN}_{\text{lin}}$ is an unusual and not very efficient way to implement linear regression, but we decided to use a neural network for the simplest (linear) case to examine a model from a model class which can be extended to general cases without further ado (only by changing the architecture). We compare the fixed effects/weights of the problem, the variance components, the prediction quality, and the different estimation times for the algorithms.

In the non-linear case, we first estimate two LME models. The first model uses only the original (linear) features (LME$_{\text{lin}}$), the second (non-linear[1]) LME knows about the simulated quadratic terms and they are included as input (LME$_{\text{nlin}}$). The latter will serve as a baseline (in the ground truth sense) for flexible machine learning predictors. Here only the original features are used and the question at hand is if the non-linear relationship can be modeled. The flexible `ml` models in the non-linear case are a 3-layer feed-forward neural network (NN$_{\text{nlin}}$ - with 5 and 3 hidden units) and a Support Vector Regressor (SVR with a radial basis function kernel).

The model specifications are shown in Table 2.

Table 2: The different models considered for the evaluations.

| | |
|---|---|
| LME$_{\text{lin}}$ | Linear Mixed Effects model with $X\beta$ |
| LME$_{\text{nlin}}$ | Linear Mixed Effects model where the specified non-linear terms are included in $X$ (we know the exact (simulated) non-linear relationship, see (12)) |
| | We use MixedLM() from statsmodels with the conjugate gradient optimizer |
| `ml`: | |
| NN$_{\text{lin}}$ | 0-hidden-layer neural network with linear output activation as the linear regression special case with Adam optimizer ($lr = 0.01$). |
| NN$_{\text{nlin}}$ | 2-layer neural network with 5 and 3 hidden units (4 and 4 units for the `hot`). We specified ReLU activation for the hidden units and a linear output activation and used tensorflow-keras for the implementation with Adam optimizer ($lr = 0.01$) and mean-squared-error loss. |
| SVR | Support Vector Regressor implemented in sklearn (for hyperparameters see Appendix C) |
| pure ml$^{\text{Est}}$ | mixedML implementation of the EM algorithm (Est: MLE or REML) or the out-of-the-box approach (Est: OOB, from statsmodels) with ml as the ml$_{\text{fixed}(\theta)}$ model. |

In order to obtain measures of the stability of the parameter estimates, we performed a $50-$fold cross-validation for all models, that means in every fold we estimate the model on a training set of a random 98%-2% split and evaluate it on the respective test set. We repeat this procedure 50 times so that every observation is exactly once in the test set, then means and standard deviations are reported. Run times for single folds are reported with the results (2,8 GHz Quad-Core Intel Core i7). For neural networks, we implemented the convergence criteria of loss differences $< 0.1$ for 8 consecutive epochs. As mentioned earlier, for the EM algorithm, we use the same procedure based on the GLL (9) for all models.

### 5.3 Results

See Table 3 for the results of the linear problem. The estimated parameters and variances of the models are in line with the traditional model (LME), indicating an unbiased parameter estimation. In terms of prediction quality, all models and estimation methods show similar results on the training and test data where a significant increase in prediction quality can be noted when the information about the cluster membership is used (*within*) compared to observation from unknown clusters. In the pure machine learning models (`ml`), the cluster membership cannot be used, so there is no *within* score comparison. The *out* scores are in line with the pure machine learning prediction results. The fixed effects machine learning approach (`hot`) should be compared to the train- and test- *within* scores, since the cluster membership is used as a fixed effects. For both scores, we can see a decrease in the prediction quality of the `hot` model compared to the other models. Even compared to the pure machine learning models (`ml`), the one-hot encoded cluster membership only leads to a little increase in quality. While parameter estimates and prediction qualities seem to be equivalent for the three estimation methods, the estimation time (for one fold) differs dramatically, with the biggest mean difference between the out-of-the-box (OOB) estimation and the MLE/REML procedures ($\Delta t = 233.62s$ (MLE) and $\Delta t = 228.87s$ (REML)). Note that the estimation times for our implemented MLE/REML estimations are also faster than the traditional LME estimation time.

The non-linear results are reported in Table 4. All parameter estimates are stable, but two main differences can be seen. When the linear traditional model (LME$_{\text{lin}}$) is compared to the traditional model where

---

[1]We refer to this model as non-linear to distinguish it from the linear LME and because the non-linear terms are considered as features. Using those features, the estimated relationship, and the model itself is nevertheless linear.

Table 3: Results for the linear problem (data).

| | sim | $\mathrm{LME_{lin}}$ | $\mathrm{NN_{lin}^{OOB}}$ | $\mathrm{NN_{lin}^{MLE}}$ | $\mathrm{NN_{lin}^{REML}}$ | $\mathrm{NN_{lin}^{ml}}$ | $\mathrm{NN_{lin}^{hot}}$ |
|---|---|---|---|---|---|---|---|
| **Fixed effects** | | | | | | | |
| bias | 5.00 | 4.89 (0.01) | 4.89 (0.03) | 4.88 (0.05) | 4.89 (0.04) | . | . |
| $x_{1,1}$ | 1.00 | 0.97 (0.01) | 0.98 (0.03) | 0.99 (0.05) | 0.97 (0.04) | . | . |
| $x_{1,2}$ | 1.00 | 0.99 (0.01) | 1.00 (0.04) | 1.00 (0.06) | 1.00 (0.05) | . | . |
| $x_{1,3}$ | 1.00 | 0.98 (0.01) | 0.98 (0.03) | 0.98 (0.05) | 0.98 (0.05) | . | . |
| $x_{1,4}$ | 1.00 | 0.96 (0.01) | 0.96 (0.03) | 0.96 (0.05) | 0.96 (0.05) | . | . |
| $x_{1,5}$ | 1.00 | 1.03 (0.01) | 1.04 (0.03) | 1.03 (0.06) | 1.03 (0.06) | . | . |
| $x_{1,6}$ | 1.00 | 0.88 (0.01) | 0.87 (0.03) | 0.88 (0.05) | 0.88 (0.06) | . | . |
| $x_{1,7}$ | 1.00 | 0.89 (0.01) | 0.89 (0.03) | 0.88 (0.04) | 0.90 (0.04) | . | . |
| $x_{1,8}$ | 1.00 | 1.01 (0.01) | 1.01 (0.03) | 1.01 (0.05) | 1.01 (0.04) | . | . |
| $x_{1,9}$ | 1.00 | 0.96 (0.01) | 0.97 (0.03) | 0.95 (0.05) | 0.94 (0.05) | . | . |
| $x_{1,10}$ | 1.00 | 1.06 (0.01) | 1.06 (0.03) | 1.07 (0.05) | 1.07 (0.05) | . | . |
| $x_{2,1}$ | 1.00 | 0.85 (0.01) | 0.85 (0.04) | 0.84 (0.04) | 0.86 (0.05) | . | . |
| $x_{2,2}$ | 1.00 | 1.10 (0.01) | 1.10 (0.03) | 1.09 (0.04) | 1.10 (0.04) | . | . |
| $x_{2,3}$ | 1.00 | 1.02 (0.01) | 1.02 (0.03) | 1.02 (0.04) | 1.02 (0.04) | . | . |
| **Variance components** | | | | | | | |
| Var(bias) | 1.00 | 1.06 (0.02) | 1.06 (0.02) | 1.04 (0.02) | 1.07 (0.03) | . | . |
| $\mathrm{Var}(x_1)$ | 1.00 | 0.78 (0.02) | 0.78 (0.02) | 0.78 (0.02) | 0.79 (0.02) | . | . |
| $\mathrm{Var}(x_2)$ | 1.00 | 1.05 (0.02) | 1.05 (0.02) | 1.05 (0.02) | 1.06 (0.02) | . | . |
| $\mathrm{Var}(x_3)$ | 1.00 | 0.89 (0.02) | 0.89 (0.02) | 0.89 (0.02) | 0.90 (0.02) | . | . |
| $\mathrm{Var}(x_4)$ | 1.00 | 1.08 (0.03) | 1.08 (0.03) | 1.07 (0.03) | 1.08 (0.03) | . | . |
| $\mathrm{Var}(x_5)$ | 1.00 | 0.90 (0.03) | 0.90 (0.03) | 0.93 (0.02) | 0.94 (0.02) | . | . |
| $\mathrm{Var}(x_6)$ | 1.00 | 1.04 (0.03) | 1.04 (0.03) | 1.06 (0.03) | 1.07 (0.03) | . | . |
| $\mathrm{Var}(x_7)$ | 1.00 | 0.84 (0.02) | 0.84 (0.02) | 0.84 (0.02) | 0.84 (0.03) | . | . |
| $\mathrm{Var}(x_8)$ | 1.00 | 0.84 (0.03) | 0.84 (0.03) | 0.86 (0.03) | 0.87 (0.02) | . | . |
| $\mathrm{Var}(x_9)$ | 1.00 | 1.20 (0.03) | 1.20 (0.03) | 1.21 (0.03) | 1.22 (0.03) | . | . |
| $\mathrm{Var}(x_{10})$ | 1.00 | 1.24 (0.03) | 1.24 (0.03) | 1.23 (0.03) | 1.25 (0.03) | . | . |
| Cov (mean) | 0.00 | 0.01 (0.00) | 0.01 (0.02) | 0.01 (0.02) | 0.01 (0.02) | . | . |
| $\sigma_\epsilon$ | 3.50 | 3.51 (0.01) | 3.51 (0.20) | 3.50 (0.19) | 3.50 (0.20) | . | . |
| **Prediction** | | | | | | | |
| $\mathrm{RMSE_{within}}$ (test) | . | 4.04 (0.18) | 4.04 (0.18) | 4.04 (0.19) | 4.04 (0.18) | 4.80 (0.28) | 4.78 (0.24) |
| $\mathrm{RMSE_{out}}$ (test) | . | 4.80 (0.28) | 4.80 (0.28) | 4.80 (0.28) | 4.80 (0.28) | 4.80 (0.28) | . |
| time (s) | . | 43.10 (3.85) | 266.81 (18.34) | 33.19 (2.28) | 37.94 (3.59) | . | . |

Notes: $x_{1/2,p}$: coefficients ($\hat{\beta}$) for the respective features of level 1/2, var(.): estimated variances of the $(p_1 + 1)$ random effects, $\sigma_\epsilon$ residual standard deviation, train/test sets according to 50-fold cross-validation (means and standard deviation), root mean squared error (RMSE) subscripts: *within*: with the corrections of the random effect realizations known for that cluster, *out*: ignoring the cluster (out of sample), *ml* only the pure ml model, *hot* the pure ml model with one-hot cluster-membership, time: estimation time for one fold in seconds.

interactions are included ($\mathrm{LME_{nlin}}$), we can see that the estimated variance components fit the simulated parameters (see Table 3) better for $\mathrm{LME_{nlin}}$, easiest to see for the $\sigma_\epsilon$ estimates, where the linear model overestimates the parameter. Interestingly, the non-linear neural network ($\mathrm{NN_{nlin}}$) seems to be equally unable to capture the non-linearity as $\mathrm{LME_{lin}}$. Here the $\mathrm{NN_{nlin}^{OOB}}$ estimates are in line with $\mathrm{LME_{lin}}$ (note that the estimation method within each EM iteration is the same here). The estimates of the same model but with the MLE/REML estimation indicate only a slightly better fit. For the SVR model, the parameter estimates are in line with the baseline model $\mathrm{LME_{nlin}}$, again with very few advantages for MLE/REML.

Those findings are also reflected in the prediction quality. Again it can be seen, that the neural network is on the level of the linear LME. The SVR prediction results can be compared with the baseline model (ground truth). Note that in the baseline $\mathrm{LME_{nlin}}$, all simulated non-linearities (interactions) are specified, whereas the SVR model only uses the original features as input and then models the non-linearity. At least in the sense of prediction, a precise model specification is thus not necessary. For the estimation times, we can see again, that our MLE/REML estimation is significantly faster than the out-of-the-box implementation.

Table 4: Results for the non-linear problem (data).

| | $\mathrm{LME_{lin}}$ | $\mathrm{LME_{nlin}}$ | $\mathrm{NN_{nlin}^{OOB}}$ | $\mathrm{NN_{nlin}^{MLE}}$ | $\mathrm{NN_{nlin}^{REML}}$ | $\mathrm{SVR^{OOB}}$ | $\mathrm{SVR^{MLE}}$ | $\mathrm{SVR^{REML}}$ |
|---|---|---|---|---|---|---|---|---|
| $\mathrm{Var}(x_0)$ | 0.62 (0.03) | 0.87 (0.02) | 0.62 (0.03) | 0.71 (0.06) | 0.74 (0.09) | 0.67 (0.03) | 0.75 (0.04) | 0.78 (0.04) |
| $\mathrm{Var}(x_1)$ | 1.43 (0.04) | 1.09 (0.03) | 1.43 (0.04) | 1.45 (0.05) | 1.46 (0.07) | 1.21 (0.03) | 1.22 (0.04) | 1.23 (0.04) |
| $\mathrm{Var}(x_2)$ | 1.47 (0.04) | 1.07 (0.03) | 1.47 (0.04) | 1.47 (0.04) | 1.48 (0.04) | 1.23 (0.03) | 1.24 (0.03) | 1.25 (0.03) |
| $\mathrm{Var}(x_3)$ | 0.57 (0.04) | 0.59 (0.02) | 0.57 (0.04) | 0.71 (0.03) | 0.72 (0.04) | 0.57 (0.03) | 0.65 (0.02) | 0.65 (0.02) |
| $\mathrm{Var}(x_4)$ | 1.26 (0.04) | 0.94 (0.03) | 1.26 (0.04) | 1.34 (0.04) | 1.35 (0.04) | 1.19 (0.03) | 1.25 (0.03) | 1.26 (0.03) |
| $\mathrm{Var}(x_5)$ | 1.14 (0.04) | 1.01 (0.03) | 1.14 (0.04) | 1.19 (0.04) | 1.20 (0.05) | 1.11 (0.03) | 1.14 (0.03) | 1.15 (0.03) |
| $\mathrm{Var}(x_6)$ | 0.79 (0.04) | 0.79 (0.03) | 0.79 (0.04) | 0.86 (0.04) | 0.87 (0.04) | 0.77 (0.04) | 0.82 (0.03) | 0.83 (0.03) |
| $\mathrm{Var}(x_7)$ | 1.01 (0.04) | 1.03 (0.03) | 1.01 (0.04) | 1.05 (0.04) | 1.06 (0.04) | 1.03 (0.03) | 1.09 (0.04) | 1.10 (0.04) |
| $\mathrm{Var}(x_8)$ | 0.93 (0.04) | 0.85 (0.03) | 0.93 (0.04) | 1.05 (0.04) | 1.06 (0.04) | 1.01 (0.03) | 1.08 (0.03) | 1.10 (0.03) |
| $\mathrm{Var}(x_9)$ | 0.86 (0.04) | 0.87 (0.03) | 0.86 (0.04) | 1.00 (0.04) | 1.01 (0.04) | 0.95 (0.03) | 1.05 (0.03) | 1.06 (0.03) |
| $\mathrm{Var}(x_{10})$ | 1.11 (0.04) | 0.96 (0.03) | 1.11 (0.04) | 1.18 (0.04) | 1.19 (0.05) | 1.04 (0.04) | 1.08 (0.04) | 1.09 (0.04) |
| Cov (mean) | -0.01 (0.00) | -0.00 (0.00) | -0.01 (0.03) | -0.00 (0.03) | -0.00 (0.03) | -0.01 (0.02) | -0.01 (0.02) | -0.01 (0.02) |
| $\sigma_\epsilon$ | 4.68 (0.01) | 3.56 (0.01) | 4.68 (0.29) | 4.65 (0.32) | 4.65 (0.32) | 4.10 (0.27) | 4.08 (0.26) | 4.08 (0.26) |
| time (s) | 48.45 (3.85) | 46.11 (5.61) | 287.74 (20.38) | 42.47 (2.92) | 44.34 (2.90) | 256.97 (5.58) | 82.51 (1.52) | 86.48 (2.94) |

Notes: $x_p$: coefficients ($\hat{\beta}$) for the respective features, var(.): estimated variances of the ($p_1 + 1$) random effects $\hat{\sigma}_\nu^2$, $\sigma_\epsilon$ residual standard deviation, train/test sets according to 50-fold cross-validation (means and standard deviation), time: estimation time for one fold in seconds.

Table 5: Prediction results for the non-linear problem (data).

| | $\text{RMSE}_{\text{within}}$ | $\text{RMSE}_{\text{out}}$ | $\text{RMSE}_{\text{within}}^{\text{test}}$ | $\text{RMSE}_{\text{out}}^{\text{test}}$ |
|---|---|---|---|---|
| **LME** | | | | |
| ·lin | 4.25 (0.01) | 5.72 (0.01) | 5.39 (0.39) | 5.72 (0.45) |
| ·nlin | 3.15 (0.01) | 4.75 (0.01) | 4.09 (0.26) | 4.75 (0.33) |
| **NN$_{\text{nlin}}$** | | | | |
| .OOB | 4.27 (0.02) | 5.74 (0.01) | 5.42 (0.39) | 5.74 (0.45) |
| .MLE | 4.21 (0.01) | 5.75 (0.04) | 5.39 (0.39) | 5.75 (0.48) |
| .REML | 4.20 (0.01) | 5.75 (0.04) | 5.39 (0.39) | 5.74 (0.47) |
| .ml | 5.74 (0.02) | 5.74 (0.02) | 5.74 (0.45) | 5.74 (0.45) |
| .hot | 5.59 (0.03) | . | 5.76 (0.44) | . |
| **SVR** | | | | |
| .OOB | 3.68 (0.01) | 5.22 (0.01) | 4.75 (0.30) | 5.22 (0.36) |
| .MLE | 3.65 (0.01) | 5.24 (0.01) | 4.75 (0.30) | 5.25 (0.37) |
| .REML | 3.65 (0.01) | 5.24 (0.01) | 4.75 (0.30) | 5.25 (0.37) |
| .ml | 5.24 (0.01) | 5.24 (0.01) | 5.25 (0.37) | 5.25 (0.37) |
| .hot | 5.17 (0.01) | . | 5.23 (0.38) | . |

Notes: train/test sets according to 50-fold cross-validation (means and standard deviation), root mean squared error (RMSE) subscripts: *within*: with the corrections of the random effect realizations known for that cluster, *out*: ignoring the cluster (out of sample), *ml* only the pure ml model, *hot* the pure ml model with one-hot cluster-membership, time: estimation time for one fold in seconds.

## 6 Empirical evaluation

In order to test the framework on real data, we use a data set introduced in the textbook Snijders & Bosker (2011) (with further reference to e.g. Brandsma & Knuver, 1989; Knuver & Brandsma, 1993). The data contains the result of a language test (target variable) of 3758 students grouped in 211 different classes/school (one class per school). Further information on the student level (level 1) is available, where we use the students' IQ and their socioeconomic status (SES). On the school level, we use the school mean IQ (sIQ), and the school mean SES (sSES). See also within- and between-group regression, for example, in Snijders & Bosker (2011) for this particular example. We use the same data set with the same preprocessing as in the textbook, meaning that IQ and SES are centered (before missings were removed, which leads to a small shift in the centering), then the school means were calculated.

### 6.1 Models and procedure

Using this data, we model the relationship of the results of the language test dependent on the students' IQ and SES with the IQ and SES school means (sIQ and sSES) as level 2 variables. We use random slopes (IQ and SES) and a random intercept. Additionally, we expect interactions between all of the variables. Note that this is a non-linear relationship between the four single predictors and the target. If all interactions are first calculated and then used as predictors, the relationship becomes linear

$$y_{ij} = \beta_0 + \beta_1 IQ_{ij} + \beta_2 SES_{ij} + \beta_3 sIQ_j + \beta_4 sSES_j + \beta_5(SES_{ij} \times IQ_{ij}) + \beta_6(SES_{ij} \times sIQ_j)$$
$$+ \beta_7(SES_{ij} \times sSES_j) + \beta_8(sIQ_j \times sSES_j) + \beta_9(IQ_{i,j} \times sIQ_j) + \beta_{10}(IQ_j \times sSES_j) + Z_j u_j + \epsilon_{ij}.$$

For this linear relationship, where interactions are specified in the model, we use a single layer feed-forward neural network as $\texttt{ml}_{\text{fixed}(\theta)}$. For the non-linear approach, where interactions have not been specified in advance, we use a Support Vector Regressor (SVR)

$$y_{ij} = \texttt{ml}_{\text{fixed}(\theta)}(IQ_{ij}, SES_{ij}, sIQj, sSES_j) + \boldsymbol{Z}_j \boldsymbol{v}_j + \epsilon_{ij}.$$

For both models, we apply the REML estimation. Note that the matrix $\boldsymbol{Z}$ is the same for both scenarios, with

$$Z_j = \begin{pmatrix} 1 & IQ_{1j} & ses_{1j} \\ 1 & IQ_{2j} & ses_{2j} \\ ... & ... & ... \\ 1 & IQ_{n_jj} & ses_{n_jj} \end{pmatrix}. \tag{13}$$

Table 6: Results of a mixedML model with a single layer feed-forward neural network (linear) and all interactions included as predictors.

|  | book results | Est. (sd) |
|---|---|---|
| **fixed effects** | | |
| int | 41.63 (0.26) | 41.67 (0.26) |
| SES | 0.17 (0.012) | 0.17 (0.02) |
| IQ | 2,23 (0.06) | 2.24 (0.06) |
| sSES | -0.09 (0.04) | -0.10 (0.06) |
| sIQ | 0.82 (0.31) | 0.92 (0.07) |
| SES x IQ | -0.02 (0.01) | -0.02 (0.01) |
| SES x sSES | 0.00 (0.00) | 0.00 (0.01) |
| SES x sIQ | 0.02 (0.02) | 0.03 (0.02) |
| sSES x sIQ | -0.13 (0.04) | -0.14 (0.07) |
| sSES x IQ | 0.00 (0.01) | 0.00 (0.02) |
| IQ x sIQ | -0.8 (0.08) | -0.06 (0.04) |
| **variance components** | | |
| Var(int) | 8.34 (1.41) | 9.31 (0.83) |
| Var(SES) | 0.00 (0.00) | 0.01 (0.00) |
| Var(IQ) | 0.17 (0.07) | 0.42 (0.05) |
| Cov(int, SES) | 0.00 (0.00) | -0.01 (0.03) |
| Cov(int, IQ) | -0.94 (0.20) | -0.91 (0.10) |
| Cov(SES, IQ) | 0.00 | -0.02 (0.01) |
| $\sigma_\epsilon$ | 6.11 (0.95) | 6.07 (0.48) |

Notes: SES: socioeconomic status, sSES/sIQ school means, × denotes interactions that were included as predictors. The variance components refer to the variances and covariances of the random effects. The REML-EM algorithm was used for estimation. $\sigma_\epsilon$ residual standard deviation, book results in Table 5.3 inSnijders & Bosker (2011)

Table 7: Prediction results for the empirical data.

|  | $\text{RMSE}_{\text{within}}$ | $\text{RMSE}_{\text{out}}$ | $\text{RMSE}_{\text{within}}^{\text{test}}$ | $\text{RMSE}_{\text{out}}^{\text{test}}$ |
|---|---|---|---|---|
| $\text{NN}_{\text{lin}}$ | 5.82 (0.03) | 6.90 (0.12) | 6.31 (0.53) | 6.88 (0.58) |
| $\text{NN}_{\text{lin}}$ (ml) | 6.84 (0.08) | 6.84 (0.08) | 6.88 (0.58) | 6.88 (0.58) |
| $\text{NN}_{\text{lin}}$ (hot) | 6.05 (0.04) | . | 6.38 (0.52) | . |
| $\text{SVR}^{\text{REML}}$ | 5.80 (0.01) | 7.09 (0.03) | 6.31 (0.54) | 7.08 (0.69) |
| $\text{SVR}^{\text{REML}}$ (ml) | 6.77 (0.01) | 6.77 (0.01) | 7.08 (0.69) | 7.08 (0.69) |
| $\text{SVR}^{\text{REML}}$ (hot) | 6.44 (0.01) | . | 6.55 (0.62) | . |

Notes: $\text{NN}_{\text{lin}}$ - interactions are specified, $\text{SVR}^{\text{REML}}$ - only original features, train/test sets according to 50-fold cross-validation (means and standard deviation), root mean squared error (RMSE) subscripts: *within*: with the corrections of the random effect realizations known for that cluster, *out*: ignoring the cluster (out of sample), *ml* only the pure ml model, *hot* the pure ml model with one-hot cluster-membership

Again, hyperparameters of the pure machine learning models were tuned separately (Appendix C) and we then perform 50−fold cross-validation to report means and standard deviations.

## 6.2 Empirical results

Parameter estimates for this model are shown in Table 6 compared to the results in Snijders & Bosker (2011) (Table 5.3). From this comparison, we again conclude that our REML estimation method is in line with traditional methods, even for real empirical data.

Since we don't get the individual parameters (of the features and their interaction) in the SVR model, we will use the predictive quality as the metric for comparison. The question here relates to whether a more flexible machine learning model incorporates information into a prediction that would otherwise have to be specified directly (e.g., as interactions). The prediction results are shown in Table 7. Here we can see that the SVR can produce the same prediction results, even though the non-linearities (interactions) are not specified. Note also, that the SVR here suffers from overfitting (when the train and test results are compared), which should be avoided by hyperparameter tuning, which was not included in our analysis.

# 7    Conclusion and discussion

With this paper, we aimed to, first, motivate to include random effects due to the chance of increased predictive power and inference. To reach this goal, we combined existing approaches into a more general and flexible framework, which does not limit the choice of machine learning procedures that can be used. Here, every machine learning predictor can be specified outside of the framework and can then be used for the fixed effects. The framework allows to specify machine learning model-specific convergence criteria, which is highly recommended since the fine-tuning of the model in every EM iteration reaches the convergence fast.

The second goal was to include different estimation procedures, especially adding the REML estimation, and compare parameter estimates and the predictive quality in a simulation study. The results show that in the linear case, the framework is able to produce the same parameter estimates as in traditional LME models, but decrease estimation time. Our MLE/REML estimations decrease the estimation time compared to the often-used OOB implementation significantly and also decreases the time compared to the traditional LME method. The lower estimation time of MLE/REML compared to OOB can only be explained by the applied variance and covariance estimation, since for both methods the machine learning fixed effects estimation is the same. Whereas in MLE/REML only the variance components are estimated, the OOB estimates a full LME model. Here random effects and fixed effects are estimated, even though the fixed effect estimation was already performed by the machine learning procedure and thus are zero in the remaining model. As a result, the OOB fixed effects estimates are consequentially zero, but nevertheless, they had to be estimated, since an empty (no fixed effects) model can not be specified. This might lead to a significant and unnecessary increase in estimation time. In the non-linear case, the framework shows good prediction results even if non-linearities like interactions are not specified. This is a big advantage since correct model specifications are almost always impossible and the robustness of procedures is desirable. In our analyses, as expected, there is little difference between the MLE and REML estimates. Nevertheless, we can and would generally recommend the REML estimator as it is robust to the number of clusters, as described in section 2.1.

As the third goal, the performance should be evaluated not only on simulated data but also on a real data set. Here the parameter estimates are in line with the traditional estimates, indicating an unbiased estimation. Furthermore, the same prediction results of a model where interactions are specified can be achieved without the need to overspecify a model or find the right interactions and non-linearities by hand, but by using a flexible framework for the fixed effects.

## 7.1    Limitations and future work

Relevant open questions should be investigated in further, detailed simulation studies. This refers especially to the area of sample size, both general sample size and different combinations regarding the number of clusters and included observations. Open questions in this area refer to the efficiency of the estimation in regards to the dependence of the data size, as well as a possible dependence (especially of the comparison models like `hot`) on different cluster sizes (e.g. underrepresented clusters). In our studies, although no balanced design was chosen, the cluster sizes are nevertheless in the same order of magnitude. The dependency on sample size also becomes relevant in the combination of a large number of clusters, with a in general small sample size. Here the fixed effects approach (here as `hot`) might suffer from a mismatch in parameters to be estimated and available observations. Another point that should be investigated in these studies is the convergence of the EM algorithm and to what extent this depends on different simulated scenarios.

Since the framework decreases estimation time, even when compared to the traditional models, it would be interesting to investigate if this advantage further increases with increasing sample size. Then the mixedML specification that intends to mimic a LME ($NN_{lin}$) can be an alternative for the psychometric community to estimate linear mixed effect models for big data sets. Further, the option of predicting the random effects/correction for unobserved clusters as in Xiong et al. (2019) should be included in a general framework. The next steps will be to implement the generalized mixed effects model allowing for link functions to non-normal target distributions (see Ngufor et al., 2019). This will allow for different output activation functions, thus including classification problems.

**Acknowledgments**

This work was supported by the German Research Foundation (EXC 2064, project number 390727645) – Cluster of Excellence "Machine Learning – New Perspectives for Science".

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

## Appendix

## A    out-of-the-box implementation - Algorithm B

OOB Algorithm for the implementation with standard mixed model methods (out-of-the-box)

| **Algorithm:** OOB implementation |
|---|
| $t = 0$      Initialize $\hat{\boldsymbol{u}}_{j(0)} = 0, \forall j = 1, ..., J$, $\boldsymbol{y}_{(0)}^{fixed} = \boldsymbol{y}$ 
 and train $\mathtt{ml}_{\mathrm{fixed}(\theta)}(\boldsymbol{X}) = \boldsymbol{y}_{(0)}^{\mathrm{fixed}} \to \hat{\boldsymbol{y}}_{(0)}^{\mathrm{fixed}}$, set $\boldsymbol{e}_{(0)} = \boldsymbol{y} - \hat{\boldsymbol{y}}_{(0)}^{\mathrm{fixed}}$ 
 $t = t + 1$    specify the LME as $$\boldsymbol{e}_{(t-1)} = \boldsymbol{X}\boldsymbol{\beta} + \boldsymbol{Z}\boldsymbol{\nu}$$ to be trained with a mixed model framework (out of the box) 
 $\to \hat{\boldsymbol{u}}_{(t)}$, $\hat{\boldsymbol{\Sigma}}_{\nu(t)}$ and $\hat{\sigma}^2_{\epsilon(t)}$ 
 Update $\boldsymbol{y}_{(t)}^{\mathrm{fixed}} = \boldsymbol{y} - \boldsymbol{Z}\hat{\boldsymbol{u}}_{(t)}$ 
 train $\mathtt{ml}_{\mathrm{fixed}(\theta)}(\boldsymbol{X}) = \boldsymbol{y}_{(t)}^{\mathrm{fixed}} \to \hat{\boldsymbol{\theta}}_{(t)}$ and $\hat{\boldsymbol{y}}_{(t)}^{\mathrm{fixed}}$ 
 set $\boldsymbol{e}_{(t)} = \boldsymbol{y} - \hat{\boldsymbol{y}}_{(t)}^{\mathrm{fixed}}$ |

## B    Simulation details

To evaluate and compare the models, we conducted a simulation study. We fix the number of clusters to $J = 200$, the number of observations per cluster $n_j$ to a sampled value between 22 and 25, the number of level 1 predictors (attributes, features) $p_1 = 10$ and the number of level 2 predictors to $p_2 = 3$. We include random effects for all level 1 predictors ($p_r = p_1$).

We simulate the level 1 feature vector for every observation $i \in [1, ..., N]$ as a random draw from the $p_1$-dimensional multivariate normal distribution $\mathcal{N}(\mathbf{0}, \boldsymbol{\Sigma}_{1x})$ with

$$\mathrm{diag}(\boldsymbol{\Sigma}_{1x}) = \mathbf{1}$$

and covariance values of 0.1 or 0.3 according to a random draw of the uniform distribution $\mathcal{U}_{0,1}$ (with threshold 0.5). Note that these chosen covariance values correspond to typical low and medium effect sizes that are usually found in the social and behavioral sciences (Cohen, 2013).

We repeat this process for the level 2 feature vectors with the according $p_2$ dimensional covariance matrix $\boldsymbol{\Sigma}_{2x}$. Note that level 2 features are equal for observations within the same cluster.

For the covariance matrix $\Sigma_\nu$ of the $p_r + 1$ random effects ($p_r$ slopes/weights plus random intercept/bias), we assume $\sigma_{\nu_m, \nu_n} = 0$ for $n \neq m$. For the then diagonal covariance matrix of the random effects, we assume the identity matrix (setting $\sigma^2_{\nu_m} = 1, \forall m \in [0, 1, ..., p_r]$). By setting the variance of the random effects to one, we identify the latent variable vector. To simulate the cluster specific realizations $\boldsymbol{u}_j$ of the latent variables we draw samples $\boldsymbol{u}_j \sim \mathcal{N}(\mathbf{0}, \Sigma_\nu)$ for $j = 1, ..., J$ and define $\boldsymbol{u} = (\boldsymbol{u}_1^t, ..., \boldsymbol{u}_j^t)^t$.

For the residuals we assume $\Sigma_\epsilon = \sigma_\epsilon \mathbf{1}$ with $\sigma_\epsilon = 3.5$, and sample $\boldsymbol{\epsilon} \sim \mathcal{N}(\mathbf{0}, \Sigma_\epsilon)$ ($\boldsymbol{\epsilon} \in \mathbb{R}^N$).

We set the intercept/bias $\beta_0 = 5$ and all other parameters to 1 and obtain $\boldsymbol{\beta}_1 \in \mathbb{R}^{p_1}$ (level 1 parameters/weights), $\boldsymbol{\beta}_2 \in \mathbb{R}^{p_2}$ (level 2 parameters/weights). With the chosen values, together with $\sigma_\epsilon$, we achieve an overall explained variances of $y$ of around 0.77 (linear) and .82 (non-linear).

**Calculate the target $\boldsymbol{y}$**    Putting it together, we obtain the linear case

$$\boldsymbol{y}_{\mathrm{lin}} = \beta_0 + \boldsymbol{X}\boldsymbol{\beta} + \boldsymbol{Z}\boldsymbol{u} + \boldsymbol{\epsilon}.$$

For the non-linear case, we include the quadratic terms of the first five level 1 features and define

$$\boldsymbol{X}_q = (x_{11}^2, ..., x_{15}^2),$$

and $\boldsymbol{\beta}_q \in \mathbb{R}^5$ sampled as the other coefficients. This gives

$$\boldsymbol{y}_{\mathrm{nlin}} = \beta_0 + \boldsymbol{X}\boldsymbol{\beta} + \boldsymbol{X}_q\boldsymbol{\beta}_q + \boldsymbol{Z}\boldsymbol{u} + \boldsymbol{\epsilon}.$$

## C   Hyperparameter tuning

Before being used in the mixedML framework, the hyperparameters of the pure machine learning models are tuned for the respective data sets. We performed an exhaustive grid search with a 3-fold cross-validation. For the linear simulation data ($NN_{lin}$), we only tune the learning rate. In the non-linear case ($NN_{lin}$), we also tune the number of hidden units in the two layers. For SVR we use the sklearn (Pedregosa et al., 2011) implementation which is based on libsvm Chang & Lin (2011). We tune the kernel (linear or radial basis function (rbf)), the regularization parameter C (inverse proportional to typical regularization parameters - range), and the rbf-kernel parameter gamma.

|  | $NN_{lin}$ | $NN_{lin}$ hot | $NN_{nlin}$ | $NN_{nlin}$ hot | SVR | SVR hot |
|---|---|---|---|---|---|---|
| **Simulations** | | | | | | |
| learning rate | 0.01 | 0.01 | 0.01 | 0.01 | - | - |
| hidden layers | - | - | 2 | 2 | - | - |
| units layer 1 | - | - | 5 | 4 | - | - |
| units layer 2 | - | - | 3 | 4 | - | - |
| kernel (linear/rbf) | - | - | - | - | rbf | rbf |
| C | - | - | - | - | 38 | 38 |
| gamma | - | - | - | - | 0.001 | 0.001 |
| | | | | | | |
| **Emprirical data** | | | | | | |
| learning rate | 0.01 | 0.01 | - | - | - | - |
| kernel (linear/rbf) | - | - | - | - | rbf | rbf |
| C | - | - | - | - | 36 | 84 |
| gamma | - | - | - | - | 0.001 | 0.001 |

