# OpenReview forum: "Mixed effects in machine learning – A flexible mixedML framework to add random effects to supervised machine learning regression"
_TMLR — Accepted by TMLR_

### Review · Reviewer_uHhv · 2023-01-15

**Summary Of Contributions:**

This paper introduces a general and flexible mixed-effects approach that incorporates random effects into supervised regression machine learning approaches to handle the dependency issue in clustered data. Herein, the framework has no restriction on the choice of the machine learning method. In addition, the framework shows promising results with decreased estimation time and improved prediction compared to other existing methods. The authors test their framework performance on simulated and real data.

**Audience:**

Yes

**Claims And Evidence:**

Yes

**Requested Changes:**

**Requested Changes**
- Acronyms/abbreviations: I found it confusing to switch between using the abbreviation and then using the full term and re-defining the abbreviation. When abbreviating a term, always spell out the full term the first time it appears/is used, followed by the abbreviation in parentheses ( e.g. Maximum Likelihood Estimation (MLE) ). Then any time you need to mention/use Maximum Likelihood Estimation, use only MLE.

-  Define/state what dose i.i.d and mixedML stand for.

-  Add the unit of time: There is a row for the time in many tables but never mention if it is measure in hours, minutes or seconds. E.g. use  time (s) if it is measured in seconds instead of time.

- Fix/Add space between the table and the caption/title. In some tables, there is a good space (e.g. Table 1), some not ( e.g. Table 2).

-  Use proper punctuation after an equation (i.e. add a comma or period):

    - If you continue writing after an equation (i.e. add another equation or where ...), add a comma. E.g. after eq(8), add a comma and after eq(9), add a comma.

    - Add a period if the sentence ends in a displayed equation. E.g. add a period at the end of the eq(10).

-  Make sure to use text environment when using a word (non-math symbol) inside math mode in the subscript and/or superscript. E.g  use this format  $y^{\text{fixed}}$ and $\text{NN}_{\text{lin}}^{\text{OBB}}$,

      instead of $y^{fixed}$ and $\text{NN}_{lin}^{OBB}$

-  Whenever mentioning a function or a programming language name, use {\sf ..... }.  Eg. use this format  $\sf python$,  and $\sf{ lmr()}$ instead of  python, and $ lmr()$.

-  Be consistent if you choose to hyphenate/not to hyphenate words (e.g. either use mixed-effect or mixed effect).

- If possible/applicable: run more simulations (see questions for details).

- If possible/applicable: describe how you split the data (see questions for details).

**Questions**

- If there are no computation issues (time or convergence), why the simulation seems run only one time with one sample size? It might be useful to repeat the simulation like 5 times or more with different sample sizes. Then report the average values of parameter estimation, time, ..., etc. to see if there are any properties or limitations.
- Is the data split into train/test? If yes, how did you split it?



**Strengths And Weaknesses:**

**Strengths**

- The proposed framework has the flexibility to be easily modified to use different ML methods.
- Simulation and real data results illustrate the promise of the framework.
- The writing style is good and balanced between using active and passive voice.

**Weaknesses**

- Properties and limitations of the framework not discussed.
- Some issues related to writing format and notation.

---

### Review · Reviewer_4AE4 · 2023-01-20

**Summary Of Contributions:**

In this paper the authors present a framework for learning mixed-effects models built around machine learning algorithms used to model the fixed-effects. It largely builds on prior work which developed expectation-maximization (EM) procedures for fitting the mixed effects models. Most of these works have used tree-based models as the core machine learning algorithm, while in the present work the authors allow for arbitrary machine learning algorithms to be used (including SVMs and neural networks). The authors then perform simulated experiments where they compare a variety of baselines (including models which ignore cluster structure in the data, and models which use cluster indicator-based fixed effects instead of random effects), as well as to ablated components of mixed effects models produced by their framework. These results show that, when well-specified, the mixed effects models are significantly better at predicting on new data from a previously observed, known cluster (i.e., cluster was in the training data), and that all approaches are comparable when data comes from unseen clusters. They then apply their framework to school language assessment dataset and demonstrate that they recover very similar coefficients when compared to those recovered by classical methods

**Audience:**

Yes

**Broader Impact Concerns:**

No broader impact concerns.

**Claims And Evidence:**

Yes

**Requested Changes:**

I think it would be most helpful for the authors to add more discussion surrounding the details of their framework. This would help clarify what makes the framework interesting (weakness 1) and is also important so that readers can fully understand (weakness 2.1) the framework and navigate the choice between REML and ML.

Currently, the authors do a good job of describing mixed effects models in Section 3. However, when describing the estimation framework in Section 4, I think the manuscript would benefit strongly from more discussion. Most importantly, the authors should include a discussion of what REML and ML are (including what REML stands for), and what makes them different. Methodologically, how do they differ (e.g., what are the consequences of the differences in the two projections in Section 4.1)? Are there situations in which we might expect one to be preferable to the other?

In a related vein, are there any methodological consequences of using non-tree-based machine learning models? Or is the EM estimation procedure exactly the same as with trees? As an example of useful information that the authors provide, when talking about benefits of ML and REML over the OOB version, the authors state that "the machine learning model can be learned from scratch, but the process can also be started with pre-trained models, which leads to early stops according to convergence criteria." This is helpful to know, and I was wondering if there were any convergence results that the authors could cite here (they would likely be model class specific though)? I also think the authors should reiterate why ML/REML have lower runtime than OOB when discussing the results in the experiments.

I also think the paper would benefit from a running example so that different model components can be tied to tangible concepts. This would be especially helpful for describing the difference between the "within" and "out" settings in a way that will be easy to understand for practitioners.

In my view answering these types of questions is important for the paper to deliver a complete account of the proposed method so that users can better navigate the framework.

On a minor note, I have suggestions for improving the presentation of the experimental results. Currently, Table 3 in particular is a large data dump, and it is hard to pull out important results from it. To help declutter the table, I think that performance on training data is less interesting (unless the authors can justify within the text otherwise) so it could be omitted or placed in an appendix. Within the table there is  nothing visual to help make items in the table stand out (e.g., bolding superior results). It might also be more effective to have visualizations for some of the data instead of using a table, such as a bar plot of the fixed effects estimates with error bars for example.

Additionally, I think a reader should be able to compare the performance of models (the "prediction" section of Table 3) on the test data by looking at a single row. So for the within [test] and out [test] rows, all methods should have their own column. At present, methods (such as pure ml and hot) are stratified across rows and columns. Instead, have the pure ml methods and the hot methods be their own columns. To draw out the significance of the fact that some of these models cannot make more accurate "within" predictions, the performance of all models on the within [test] data should be listed instead of blanked out. To make sure I understand correctly, couldn't I use a pure "ml" model to predict on a within-sample? The prediction won't differ from the value the model would predict if the cluster were not known (i.e., the within and out performances will the same, correct?), but the performance comparison with other methods that can account for the cluster is useful to see.

Other minor notes:

- It is confusing for "ml" and "ML" to both be used as acronyms that refer to different things
- Regarding sample sizes for the simulated experiments: I expect a priori that one-hot encoding will suffer most when sample sizes for some groups are very small. And I also expect pooling all the data (i.e., the pure ml model) will perform poorly (stratified, e.g., by worst performing cluster) when, say, one cluster is over represented in the training data. If not including simulated cases to demonstrate this, could the authors add more explicit discussion of when one-hot encoding (or more generally considering the random effects as fixed effects) might fail?
- The authors make many loose connections to transfer learning, but it isn't obvious to me. I think if the authors are going to lean into this connection, they should explain it explicitly at some point in Section 2.

**Strengths And Weaknesses:**

Strengths:
- The authors nicely describe the problem in the introduction. I wish they would expand further and use a running example to help explain the various components of their framework.
- The authors perform extensive experiments that compare against natural alternatives (e.g., one-hot encoded fixed effects).

Weaknesses (see more discussion below):
- While I understand that novelty is not a consideration for this review, and the authors are honest in stating they "do not aim to introduce a new model that competes with the mentioned research," it is not clear to me what generalizable insights the authors are claiming to contribute. The authors state they would like to "point out existing results and show that hierarchical data structures... can also be considered as an addition to one's own analyses" but unfortunately I'm not sure what this means. It would be helpful for the authors to clarify what they think a reader can takeaway from their study after reading the paper given that they are replicating previous methods (this also ties into the next weakness)
- I think there are a number of ways in which the clarity of the presentation could be improved.
  - Most importantly, some key elements of the framework are never discussed. As far as I could tell, there is no discussion of the difference between ML and REML other than the differing formulas provided in Equation 4.1. Further, I don't believe the acronym REML is defined.
  - While the experiments are thorough, I found the presentation of the results (particularly the tables) hard to parse. I have some suggestions for how the authors could improve this so that it is easier for readers to identify takeaways from experiments.

---

### Review · Reviewer_JZWV · 2023-01-20

**Summary Of Contributions:**

In this paper a new algorithm for fitting the mixed effect regression model is described and evaluated. Motivation for the approach is enabling flexibility of using arbitrary function approximator, like for example Neural Networks, to model the fixed effect patterns. The algorithm is based on Expectation-Maximization procedure, where parameters of random effects are estimated in E-step, and fixed effect part is optimized in M-step. Mixed effect regression problem is described and formally introduced, along with the derivations, and modeling assumptions on the form of random effects which allows for efficient computation of the solution, in particular uncorrelated multivariate normal distribution. The approach is empirically evaluated on linear and nonlinear synthetic examples, as well as in the application on language test grades of thousands of students, coming from few hundred different schools. The main takeaways appears to be superior computational efficiency with comparable results, agains the out of the box approach.


**Audience:**

Yes

**Claims And Evidence:**

No

**Requested Changes:**

Some suggestions, which I believe would help to convey the messages more clearly:
- Since 50 fold crossvalidation procedure was performed, I guess it wouldn't be much more effort to also perform the hyperparameter tuning of baseline ML model.
- Even though Mean Squared Error is a valid measure, the Square part is making differences more pronounced. So maybe is more fair reporting Root Mean Squared Error instead.
- Transfer learning was just mentioned in the pass, without giving more details or references on how it would be achieved, or what would be the pros/cons, etc. If its really relevant for the paper, its useful to elaborate on it more.
- Presentation is clear for the most part, and even though in general paper is reading easy, there are some sentences which require effort to decipher. For example sentence "Nested data induce similarities between observations when they belong to a cluster." sounds a bit circular because its stating that observations within a group are similar because they belong to the group. It would be more informative if it explains how the similarity is induced, or what the implications of this similarity are.
- Nonlinear mixed model formulas in section 2.2 are missing and equation numbers.

And few questions:
- Does first equation in section 2.2 implies that fixed effect mapping(s) $f_j$ are (potentially) different for every $j$?
- Why covariance matrix of random effects $\Sigma$ doesn't have subscript $j$? Isn't it possible that covariance differs among the groups.



**Strengths And Weaknesses:**

The main strength of the work is in the computational efficiency of the algorithm, which in addition to the flexibility of modeling frameworks that can be used, is making it attractive tool for modeling the data where samples tend to form the clusters. And such applications are numerous, so the potential for impact is there.

However, one of the weaknesses is that empirical evaluation doesn't seem compelling. Some of the evaluation setup choices are not justified, or at least not clearly presented. Claiming that model (hyperparameters) are not tuned because 'the focus is to compare approaches regarding the consideration of random effects' does not sound compelling. It might cast the doubt that mixed models are just capitalizing against not properly fitted baseline. Similarly, using the Neural Network to 'simulate' linear model is peculiar, and potentially affecting the computation times, as implementation of simpler model like Linear Regression is expected to be at least a bit more efficient.

---

### Decision · Action_Editors · 2023-02-23

**Recommendation:** Accept as is

**Comment:**

The authors and reviewers engaged in a lengthy discussion on the merits of the paper, claims, evidence, and presentation. The authors made several revisions to the paper during the review process as a result of those conversations and highlighted the changed that they made to the manuscript in blue text. After this discussion the muscript is much improved and the reviewers recommended that the manuscript be accepted.

**Audience:**

This paper is of interest to the TMLR audience because some factors in nonlinear models are measurable, but not controllable and an investigator may decide that such factors are best accounted for as random effects. Being able to estimate the parameters of a nonlinear model with random effects in a computationally efficient way would be beneficial to investigators to choose such a model.

**Claims And Evidence:**

The paper claims a novel mixed effects machine learning framework that includes random effects in supervised regression machine learning models. It also claims several estimation algorithms for the random effects. A benefit of the approach is that it can be applied to existing regression models and this unlike other methods in the field. The paper contains a thorough literature review to lend evidence to the claim that the method is novel and not found elsewhere in the literature. It demonstrates how it can be added to existing nonlinear models with simulation and an analysis of an existing data set. The authors show that the estimated parameters using their estimation algorithm are very similar to those provided by existing methods, but can be computed at a significantly reduced computation time.